# External Load Analysis in Beach Handball Using a Local Positioning System and Inertial Measurement Units

**DOI:** 10.3390/s22083011

**Published:** 2022-04-14

**Authors:** Carsten Müller, Christina Willberg, Lukas Reichert, Karen Zentgraf

**Affiliations:** 1Movement and Exercise Science in Sport, Institute of Sport Sciences, Goethe University Frankfurt, 60487 Frankfurt (Main), Germany; willberg@sport.uni-frankfurt.de (C.W.); reichert@sport.uni-frankfurt.de (L.R.); zentgraf@sport.uni-frankfurt.de (K.Z.); 2Department of Applied Health Sciences, Hochschule für Gesundheit, 44801 Bochum, Germany

**Keywords:** performance analysis, load analysis, player load, inertial movement analysis

## Abstract

Beach handball is a young discipline that is characterized by numerous high-intensity actions. By following up on previous work, the objective was to perform in-depth analyses evaluating external load (e.g., distance traveled, velocity, changes in direction, etc.) in beach handball players. In cross-sectional analyses, data of 69 players belonging to the German national or prospective team were analyzed during official tournaments using a local positioning system (10 Hz) and inertial measurement units (100 Hz). Statistical analyses comprised the comparison of the first and second set and the effects of age and sex (female adolescents vs. male adolescents vs. male adults) and playing position (goalkeepers, defenders, wings, specialists, and pivots) on external load measures. We found evidence for reduced external workload during the second set of the matches (*p* = 0.005, *η*_p_^2^ = 0.09), as indicated by a significantly lower player load per minute and number of changes in direction. Age/sex (*p* < 0.001, *η*_p_^2^ = 0.22) and playing position (*p* < 0.001, *η*_p_^2^ = 0.29) also had significant effects on external load. The present data comprehensively describe and analyze important external load measures in a sample of high-performing beach handball players, providing valuable information to practitioners and coaches aiming at improving athletic performance in this new sport.

## 1. Introduction

Beach handball has its origins in 1992 and has been growing ever since then. Derived from classic indoor team handball, beach handball is played predominantly during the summer break of team handball schedules. Although both sports share many characteristics, there are also substantial differences, placing new requirements on athletes and coaches in terms of physical demands, tactics, and designing appropriate training programs. In contrast to indoor team handball with seven players, beach handball is played on the sand by only four players, including the goalkeeper. In the offensive phase, player positions are specialist, wing, and pivot. During defensive phases, usually, other players take over the tactical tasks of the goalkeeper and defenders. Moreover, beach handball is played on a field measuring 27 × 12 m, and the game is divided by a five-minute break into two halves with a duration of 10 min each, followed by a shoot-out in the case where both teams win one half. Two more examples of the modified rules in beach handball are the possibility of repeatedly substituting players at any time, as well as variable scoring depending on the type of goal throws, e.g., trick shots such as spin shots or goalkeeper’s goals count double. These new requirements for athletes and their coaches have prompted further research efforts in various sports science disciplines, encompassing sports medicine [1], notational analyses [2], and sports psychology [3].

The increasing professionalization of the discipline and its public appeal makes beach handball an interesting candidate for the Olympic tournament. This potential is currently not reflected in the scientific literature, with only a small number of studies on requirements and conditional analyses in this sport. However, recent technological advances in terms of improved portability when using inertial measurement units (IMU) compared with the more established motion-analysis equipment allows for enhanced ecologically valid data collection outside the lab [4,5], which is particularly important for novel outdoor variations in classic team sports such as beach volleyball [6], 3 × 3 basketball [7], or beach handball [8].

The physical requirements in sports can be categorized as internal and external workload. External load refers to the physical work performed in training or competition [9]. Among others, the external load can be described as distance covered or velocity generated. Further possible indicators are accelerations, changes in direction, or jumps performed within a competition, which can be assessed using IMUs [10]. These inertial sensors include a gyroscope, an accelerometer, and a magnetometer and have been shown to successfully quantify accelerations and decelerations in all three orthogonal axes with acceptable validity [11]. On the other hand, internal load describes the individual and psychophysiological responses being elicited by external loads. Methods used to describe internal load are ratings of perceived exertion or heart rate [12]. Monitoring of player workload has recently received increasing interest to better understand the impact of workload in terms of athletic performance, fatigue, or injury risk [13,14].

In recent years, multi-camera video systems have been replaced by Global Positioning Systems (GPS) and Local Positioning Systems (LPS) for location tracking. Both tools avoid the drawbacks of camera-based technology such as the overlapping of players in team sports or the tracking of individuals under unfavorable light and shadow conditions [15,16,17]. As a result, previous studies analyzing physical demands in beach handball primarily focused on GPS-based tools [8,18,19,20]. However, recent validation studies demonstrated superior validity and reliability for radio-based LPS compared to GPS-based tracking systems [15,21,22]. For instance, LPS had higher accuracy than GPS for measuring the player’s position in space, while speed and acceleration errors were comparable between both systems [21]. In addition, LPS technology demonstrated superior validity and reliability overall compared to GPS and is therefore recommended for (elite) team sports [15].

Although the tracking of physical workload by means of monitoring exercise volume and intensity is likely to assist practitioners and coaches by delivering valuable information on the physical requirements in novel sports such as beach handball [18], only a few studies have conducted such analyses in this sport so far. For instance, Pueo et al. demonstrated that female and male athletes cover 1118 ± 222 m and 1235 ± 192 m on average during a single match, corresponding to a mean distance of 60 m·min^−1^ and 70 m·min^−1^, respectively [20]. Velocity averaged 3.9 km·h^−1^ in female athletes and 4.2 km·h^−1^ in male athletes [20]. Interestingly, the authors described lower mean distances covered in the second half of the matches, which is in line with the results presented by Sánchez-Sáez et al., who also pointed out differences in terms of player load between both half-times [8]. Likewise, differences in external workload measures were reported between female and male players [19,23] and between youth and adult players [23]. A recent study assessed the performance of 57 elite beach handball players during competition and confirmed the above results, but also found significant differences depending on the specific playing positions [19]. For instance, significant differences between the distance traveled and the number of accelerations and decelerations, but no differences in the total number of jumps were found [19]. This differentiated analysis has not been conducted in beach handball before. However, this approach seems reasonable to design sound and individualized training programs and recovery strategies [13,19]. Therefore, similar studies in this new sport are needed to support or challenge the above findings. Furthermore, an essential prerequisite for this differentiated analysis approach is the use of valid and reliable tracking technology for capturing movement patterns of beach handball players during official games and thus better describing the external workload structure in this novel discipline.

Therefore, the objectives of this study were to describe external workload measures in German beach handball players using IMUs and LPS. When following up on previous work in this area, in-depth analyses addressed probable workload differences between the first and the second set of the matches, between female and male players, between adolescent (under 18 years) and adult players (18 years and older), and between playing positions characteristic in beach handball.

## 2. Materials and Methods

### 2.1. Subjects

Based on cross-sectional analyses, we examined data of an ad hoc sample comprised of 69 German elite beach handball players acquired between August 2020 and August 2021. All participating athletes were part of the German national or prospective national team at the time of the measurement. In total, 34 official games were analyzed during four national tournaments. Before enrollment in the study, all players and coaching staff were informed about the objectives of the study, the procedures, as well as the risks and benefits of participation. All athletes and legal guardians of underage players gave written informed consent before data collection. The study was approved by the local ethics committee (Chair: Klein, A.; 2021-30, 28 June 2021).

### 2.2. Material

Self-reported height (cm) and body mass (kg) were assessed to describe the anthropometric characteristics of the study sample, and the body mass index (BMI, kg·m^−2^) was calculated. We assessed external load parameters using LPS (Catapult ClearSky, Catapult Sports©, Melbourne, Australia) and Vector 7 IMUs (Catapult Sports©). Previous studies demonstrated its accuracy in validation studies against infra-red camera systems as reference. For instance, a comparison with a Vicon motion analysis system using 20 cameras capturing at a frequency of 100 Hz revealed a mean root mean square error of 0.20 ± 0.05 m for inter-unit distance, indicating that ClearSky LPS can be confidently used to capture spatiotemporal variables [24]. Likewise, a comparison with a Qualisys motion analysis system using eight cameras capturing at 100 Hz resulted in a mean difference for all position estimations of 0.21 ± 0.13 m and an average difference in the distance well below 2% for a variety of tasks performed, once the setup of the LPS was arranged symmetrically [25]. To the best of our knowledge, no reliability studies specifically for Catapult ClearSky LPS are available yet, but a recent systematic review summarized that LPS technology provides a reliable way to measure distance variables and athletes’ average speed [26]. According to the manufacturer’s recommendations, we installed 20 anchor nodes covering the complete pitch. The initial LPS calibration was performed using a tachymeter (Leica TS06 Total Station, Leica Geosystems AG, St. Gallen, Switzerland). Local positioning was conducted with a frequency of 10 Hz. LPS locates Vector 7 IMUs (receiver tags) using a narrow ultra-wideband frequency (3.1 to 10.6 Hz) to track player position on the field. The Vector 7 IMU is a small (8.1 × 4.3 × 1.6 cm) and light (53 g) device, thus little intrusive during gameplay [5]. Each athlete was equipped with a device fixed into a neoprene vest (Vector Elite Vest, Catapult Sports©) and positioned at the upper thoracic spine level. The IMU includes a 3D accelerometer (±16 G), a 3D gyroscope (±2000 degrees per second), and a 3D magnetometer (±4900 μT), each capturing data at 100 Hz.

The manufacturer’s software (Openfield^TM^ version 3.3.0, Catapult Sports©) was used for positional and inertial movement analysis (IMA) data processing. Micromovements can be analyzed, regardless of unit orientation and positional data, by combining three-dimensional accelerometer and gyroscope data using Kalman filtering techniques. These are based on an optimal estimation algorithm to reduce accelerometer and gyroscope data variance to improve position estimation. The prediction of object trajectories allows for movement differentiation of the device and the athlete [7].

### 2.3. Study Variables

The parameters presented here can be categorized as positional data, captured at a sampling rate of 10 Hz, and encompassing the distance traveled (total distance (TD) in meter (m) and distance per minute (D/min, [m·min^−1^]), as well as maximum velocity (kilometers per hour, V_MAX_ [km·h^−1^]). IMA metrics, captured at a sampling rate of 100 Hz, include total player load and player load per minute, acceleration and deceleration, explosive efforts, changes in direction (CoD), and jump counts. Player load is a commonly used variable to describe external load in arbitrary units (au) that can be calculated using 3D-acceleration data based on the formula, where a equals acceleration [27]:$$\mathrm{player}\;\mathrm{load}=\sqrt{{\left(a_{x1}-a_{x-1}\right)}^{2}+{\left(a_{y1}-a_{y-1}\right)}^{2}+{\left(a_{z1}-a_{z-1}\right)}^{2}}$$

Thus, player load is a summative measure that estimates how many accelerations a player experiences and hence describes the biomechanical load of the body in all three planes over the activity period of interest [9]. IMA accelerations and decelerations reflect positive and negative acceleration values based on the manufacturer’s algorithm using IMU data. Movements exceeding 3.5 m·s^−2^ are summarized as explosive efforts, regardless of their direction. CoDs are defined as player movements within 45° to 135° to the right and movements within –45° to −135° to the left, with 0° being a straight forward movement. CoD values are presented as the sum of both directions. Movements below 5.0 km·h^−1^ were excluded from data analysis, serving as a low-velocity threshold, as were accelerations when values below 0.8 m·s^−2^ were detected. Jumps were identified using the manufacturers’ algorithm using the athletes’ vertical acceleration profile.

### 2.4. Procedures

During official beach handball tournaments, teams play several matches. A maximum of four games were played on one tournament day, and a maximum of seven matches were played during a whole tournament encompassing two days (weekend), which is in line with previous reports, e.g., [8]. Shoot-outs were required in some games to determine the winner. However, these were not analyzed. Following data collection, data sets were transferred to the OpenField Cloud (Catapult Sports©). In terms of substitutions, times outside the pitch were not included in the analyses by cutting the time periods once the players reached their starting position in the substitution area until they started moving back to the field. To be included in the final data analysis, players had to be active on the court for at least two minutes.

### 2.5. Statistical Analysis

Statistical analyses were performed using IBM SPSS v.28 (IBM Corporation, Armonk, NY, USA). Data sets were examined for outliers by initial standardization using z-transformation and subsequently identifying outliers using the interquartile range (IQR) method [28]. Values were considered outliers (val_out_) if they fulfilled one of the following conditions:val_out_+ > Q3 + k (IQR), or val_out_− < Q1 − k (IQR).

IQR represents the range between the first and third quartile (25th to 75th percentile). The constant k, used to scale the limits of the outlier tolerance field, was set at 1.5 [28]. Subgroup analyses were performed by comparing the physical demands of female adolescents (<18 years, *N* = 16), male adolescents (<18 years, *N* = 19), and male adults (≥18 years, *N* = 34), comprising subgroup 1: “age/sex” and between the five different playing positions (goalkeeper, defender, wing, specialist, and pivot, comprising subgroup 2: “position”). Multivariate analysis of variance (MANOVA) with Bonferroni adjustments for multiple comparisons was performed to test for anthropometric differences between the subgroups. Repeated-measures MANOVA with Bonferroni adjustments for multiple comparisons were conducted to test for differences in external load measures between the first and second sets of matches. Interaction effects (set × subgroup 1 and set × subgroup 2) were analyzed to evaluate the effects of the match set on external load measures as a function of age/sex and playing position. Multivariate analyses of covariance (MANCOVA) with Bonferroni adjustments for multiple comparisons were carried out to examine differences in all variables analyzed, with subgroup 1 adjusted for playing position and subgroup 2 adjusted for age/sex. Results are presented as means ± standard deviation and effect sizes (partial eta squared, *η*_p_^2^). Statistical significance was set at *p* < 0.05.

## 3. Results

Following the data cleaning, 285 data sets were valid and included in the analyses. Reasons for missing data were limited time on the pitch (<2 min, *N* = 23) and invalid data (*N* = 2). The overall sample had a mean age of 20.1 ± 4.9 years (range 15.3–34.4 years). The mean age among the subgroups was 16.6 ± 0.5 years for female adolescent athletes, 17.1 ± 0.4 years for male adolescent athletes, and 24.3 ± 4.7 years for male adult players. The distribution among the subgroups and corresponding anthropometric data are presented in Table 1. Significant differences were found in body height between female and male adolescents and between female adolescents and male adults, but not between male subgroups. Mass differed significantly between all three groups. BMI in male adults was significantly higher compared with the other two groups, which did not differ significantly in BMI. No significant differences in anthropometrics were found for the comparison of the second subgroup comprising the five different playing positions.

Mean play time during the first and second set was 6.7 ± 2.7 min and 6.6 ± 2.6 min, respectively (*p* > 0.05). Likewise, no differences among the subgroups were found in the mean play time during the first and second sets. A repeated-measures MANOVA showed a statistically significant difference between the first and the second sets of the matches played (*F*_(9250)_ = 2.70, *p* = 0.005, *η*_p_^2^ = 0.09, Wilk’s λ = 0.91). Post hoc analyses with Bonferroni adjustments indicated significant differences in player load per minute and changes in direction per minute (Table 2). No significant interaction effects between the set of the matches and the subgroups were found, indicating that the effects of the set on external player load was not significantly affected by the subgroups: set × subgroup 1 (“age/sex”): *F*_(2256)_ = 0.95, *p* = 0.388, *η*_p_^2^ = 0.01, Wilk’s λ = 0.99, and set × subgroup 2 (“playing position”): *F*_(4254)_ = 0.91, *p* = 0.457, *η*_p_^2^ = 0.01, Wilk’s λ = 0.99.

Table 2 further provides an overview of the effects of both subgroups on all variables analyzed. One-way MANCOVA revealed a significant effect of subgroup 1 on external player load parameters: *F*_(20,506)_ = 7.10, *p* < 0.001, *η*_p_^2^ = 0.22, Wilk’s λ = 0.61. Likewise, external player load was also affected by playing position: *F* _(40,954)_ = 10.38, *p* < 0.001, *η*_p_^2^ = 0.29, Wilk’s λ = 0.25.

### 3.1. Detailed Results of Positional Analyses Using LPS

Total distance traveled, distance per minute, and maximum velocity were evaluated by means of positional analyses using LPS. Overall, the distance traveled during a game was 806 ± 214 m, and 63.7 ± 14.3 m·min^−1^, with a mean decrease of 11 m and −1.2 m·min^−1^ during the second set. Male adolescents covered the greatest distances (870 ± 217 m), followed by male adults (790 ± 205 m) and female adolescents (760 ± 217 m). In terms of playing position, the rank order was specialists (889 ± 239 m), wings (823 ± 245 m), pivots (804 ± 158 m), followed by defenders (785 ± 172 m), and goalkeepers (737 ± 251 m). The distance traveled per minute was calculated to account for active playing time on the pitch, with similar trends among the subgroups (Figure 1). The whole sample completed 64.3 ± 14.3 m·min^−1^. Male adolescents (69.2 ± 17.1 m·min^−1^) covered significantly more distance compared with male adults (62.2 ± 13.0 m·min^−1^) but not compared to female adolescents (63.3 ± 12.0 m·min^−1^). Specialists traveled significantly larger distances (73.9 ± 15.0 m·min^−1^) compared to the other playing positions (Figure 1).

The average maximum velocity in this sample was 16.5 ± 2.0 km·h^−1^ with no changes between the first and second set of matches played (15.5 ± 2.1 km·h^−1^ vs. 15.5 ± 2.2 km·h^−1^). Results from pairwise comparisons indicated that male adolescents (17.3 ± 2.0 km·h^−1^) attained higher maximum velocities compared with female adolescents (15.6 ± 1.3 km·h^−1^, *p* < 0.001) and male adults (16.3 ± 2.0 km·h^−1^, *p* = 0.001), whose values did not differ significantly (*p* = 0.056). Regarding playing positions, the highest maximum velocity was measured in wings (17.3 ± 1.9 km·h^−1^), the lowest in goalkeepers (15.4 ± 2.2 km·h^−1^, *p* = 0.003). The remaining pairwise comparisons revealed no further significant differences (Appendix A).

### 3.2. Detailed Results of External Load Variables Using IMUs

The total player load was 92.8 ± 28.4 au with a modest decrease of 1.5 au from the first to second set played (−3.2%, *η*_p_^2^= 0.01). In order to account for playing time, player load per minute was calculated (Table 3). During the second set, a significantly lower player load per minute was found (*p* = 0.039). The analysis of interaction effects (set × subgroups) was not significant, indicating that this reduction was neither affected by age/sex nor by playing position. Male adolescent players attained a significantly higher player load compared with female adolescents and male adults (each *p* < 0.001), whose values did not differ significantly (*p* = 0.051). In terms of playing position, player load per minute was considerably lower in goalkeepers compared with the other positions, but pairwise comparisons did not reveal any significant differences (Appendix A).

The maximum acceleration (2.97 ± 0.40 m·s^−2^) and deceleration values (−3.34 ± 0.67 m·s^−2^) of the sample did not show significant differences between the first and second set of the matches, as was the case for absolute numbers of explosive efforts (9.7 ± 5.9 counts) and efforts per minute (0.78 ± 0.48 counts). Male adolescents attained the highest accelerations and decelerations and performed the highest number of explosive efforts per minute, followed by male adults and female adolescents. Pairwise comparisons indicated significantly higher maximum accelerations and decelerations in male adolescents compared to female adolescents and significantly higher maximum decelerations between male adolescents and male adults (Appendix A). Female adolescents and male adults only differed significantly in terms of maximum acceleration. The number of explosive efforts did not differ significantly between the three groups (Table 3, Appendix A). Regarding playing positions, the highest maximum accelerations and decelerations were attained by wings, the lowest by goalkeepers. Pairwise comparisons revealed no significant differences for maximum accelerations, but wings performed significantly higher decelerations compared with goalkeepers and specialists. Likewise, wings performed the highest number of explosive efforts per minute among the various playing positions, but significant differences were only found between wings and defenders (Table 3).

Further results of external load analyses comprise the mean number of IMA accelerations (19.0 ± 8.6 counts) and IMA decelerations (21.1 ± 9.6 counts), the average number of changes in directions (87.8 ± 36.4 counts), and the number of jumps (11.3 ± 7.2 counts) per match played. Table 4 and Figure 2 summarize these results as a function of the active time on the pitch (counts per minute).

The data indicate a reduction in external load measures from the first to the second set, but a significant difference was only found for the mean number of changes in direction per minute (7.3 ± 3.2 vs. 6.8 ± 3.0 counts, *p* < 0.001). No interaction effect (set × subgroups) was found, indicating that these reductions were neither affected by age/sex nor by playing position. The results further reveal a similar trend across subgroup 1 “age/sex” for the various variables analyzed, according to which male adolescents reveal the highest external load, followed by male adults and female adolescents, with the variable DEC_IMA_/min being the only exception (Table 4). However, pairwise comparisons did not reveal significant differences between female and male adolescents and male adults.

In-depth analyses evaluating the effect of playing positions on external load measures revealed that goalkeepers attained significantly more IMA-detected accelerations per minute compared to defenders, wings, and specialists (all *p* < 0.001) and also compared to pivots (*p* = 0.044) (Appendix A). Specialists acquired significantly fewer IMA accelerations per minute compared to wings and pivots (all *p* < 0.001), while defenders performed significantly less compared to pivots (*p* = 0.008). Contrary, IMA deceleration counts per minute were significantly higher in defenders compared to goalkeepers (*p* = 0.003), specialists (*p* < 0.001), and wings (*p* = 0.004) (Appendix A). Besides the highest IMA deceleration counts per minute, defenders also performed the most changes in direction per minute on the pitch (Figure 2b), while the remaining comparisons were not significantly different. Finally, goalkeepers performed significantly fewer jumps per minute compared with wings (*p* = 0.003), pivots, and specialists (both *p* < 0.001). The remaining comparisons were not statistically significant (Appendix A).

## 4. Discussion

The objectives of this study were to comprehensively describe important external load parameters using valid assessment methods and considering analyses presented in previous publications. In summary, only two variables (player load per minute and changes in direction per minute) revealed a significant reduction from the first to second set without interaction effects of subgroups, indicating limited evidence for reductions in external load measures during the second set of analyzed matches. Several variables point to differences between female and male players, as well as between adolescent and adult players, while the most significant differences in external load were found between playing positions. Additionally, this comparison revealed the largest effect size, indicating that among all factors analyzed, playing position had the greatest impact on external load measures.

One of the most common variables examined to describe external load in beach handball is the distance traveled. Our data show that the mean distance covered during matches equals 806 m, ranging from 760 m in female adolescents to 870 m in male adolescents. This is comparable with previous studies reporting a mean distance of 898 m per match in nine elite female athletes [8] and in line with a mean distance traveled of 740 m in 32 female adult players and 891 m in 25 male adult athletes [19]. However, our data reveal a considerable difference from the study by Pueo et al., who reported higher distances for elite male and female athletes (1235 m and 1118 m, respectively) [20]. This difference might best be explained by the active play time (~18 min), which is substantially higher compared to the play time in our present study (6.7 and 6.6 min per set) and indicative of a lack of substitutions. Furthermore, Pueo et al. excluded goalkeepers from their analyses, which may have contributed to the differences observed. However, since beach handball is characterized by unlimited substitutions, more meaningful data would be attained by considering the active playing time on the pitch.

In this regard, our results align well with the findings of Pueo et al. [20], who demonstrated relative distances of 69.7 m·min^−1^ in males and 59.8 m·min^−1^ in females, compared to 69.2 m·min^−1^ in male adolescents and 63.3 m·min^−1^ in female players in the present study. Remarkably, relative distances traveled in beach handball are much lower compared to small-sided team handball, which is played 4 × 4 on a court of comparable size (24 × 12 m). Corvino et al. reported a mean distance traveled of 118.5 m·min^−1^ [29], distinctively higher values compared with beach handball, which might best be explained by differences resulting from playing on firm and sand surfaces, as running on sand requires more mechanical work and results in higher energy expenditure [30], thus limiting the overall running performance compared to rigid surfaces.

Beach handball is further characterized by numerous high-intensity actions throughout the game. The game dynamics in beach handball emerge from the rapidly occurring transition from attack to defense and counterattack, requiring high-intensity game actions such as sprints, jumps, changes in direction, as well as high accelerations and decelerations. In order to design reasonable training programs aiming at sustainable performance optimization, valid assessments and detailed analyses of these high-intensity bouts of play are required. In this respect, our data largely support previous studies in beach handball but also expand them by conducting in-depth analyses on workload in subgroups. In summary, our data on maximum velocity well align with Sánchez-Sáez et al. [8], who used GPS to assess positional data with a sampling frequency of 15 Hz and reported a mean maximum velocity of 15.8 km·h^−1^ in female elite beach handball players, compared to 15.6 km·h^−1^ in female adolescent athletes in the present study. Zapardiel et al. [19] reported a mean of 5.7 and 2.5 jumps within a 10-min set in male and female athletes, respectively. This is slightly lower compared to the mean number of jumps in our male adult players and much lower compared to our adolescent female players. This difference was surprising, as the authors used similar IMU devices with an identical sampling frequency. Jumps were also analyzed in a sample of 125 beach handball athletes, comprising under 16- and under 18-year-old players, as well as adult female and male players in a study by Gomez-Carmona et al., who used WIMU^TM^ IMU devices [23]. They reported the largest effect sizes of sex on the number of jumps in U16 players, while these effects diminished with increasing age, converging to the differences observed between adolescent female and male players in our present study. Of note, the absolute number of jumps performed per game was considerably higher in the aforementioned study, in which the authors excluded players from their analyses who were active less than 30% of the time in each half but did not report the resulting active playing time, thus limiting the comparability between studies with regard to the number of jumps performed. Eventually, the reasons for the observed differences cannot be fully elucidated at this point but might most likely be explained by the heterogeneity of the samples and could be subject to further analyses in future studies.

Previous studies demonstrated a reduction in external load measures from the first to the second set, likely indicating fatigue effects over the course of a match. For instance, Sánchez-Sáez et al. reported significantly reduced total distance and average velocities during the second set [8]. Pueo et al. also found a decrease in total distance covered in the second half of the matches, but this only occurred in female players and was accompanied by a significant increase in the average velocity. Though most variables examined in the present study also point to reduced external load measures during the second set, a significant difference was only found for player load per minute and the number of changes in direction per minute. However, in contrast to Pueo et al. [20], we found no significant interaction effects between the set played and the subgroups analyzed. Therefore, the decline observed in external load measures was nearly equivalent among the subgroups and therefore neither affected by age/sex nor by playing position and could therefore be interpreted as an indication of potential overall fatigue effects.

With regard to the effects of sex and age on external load measures in elite beach handball players, previous studies consistently demonstrated differences between female and male players [19,20], but also between youth/ adolescent and adult players [18,19]. In principle, our results confirm these findings by pointing to consistently higher external load in males compared with female adolescent players across the various variables analyzed. Statistically significant differences were found between these two groups in half of the outcomes, namely, distance traveled per minute, maximum velocity, maximum acceleration and deceleration, and player load per minute. This finding might be best explained by sex-specific differences in physical fitness profiles among players, which consecutively translate to different amounts of high-intensity movements in beach handball. For instance, Lemos et al. reported superior performance in males compared to female elite athletes in all fitness tests analyzed, with differences ranging between 9 and 24% [31]. Tests comprised 15 m runs with 5 m distance for acceleration, 15 m distance for a sprint, as well as horizontal jumps. Interestingly, they also found that U21 and senior male players outperformed U19 male players in the horizontal jump, while 5 m acceleration and 15 m sprint were comparable among the three groups [31]. Assuming that physical fitness profiles are reasonably comparable between adolescent and adult players and can be translated to performance in the game, further factors must be considered that explain our results, according to which adolescent players reveal higher external load in nine out of ten variables with significant differences found for the distance traveled per minute, maximum velocity, player load per minute, and maximum acceleration. Gomez-Carmona et al. argued that the anthropometric profile has a direct impact on external workload in beach handball, based on their observation of a positive association between load measures (e.g., player load, jumps per minute) and BMI values [23]. Likewise, BMI of adult players was higher in our present study compared to adolescent players. We also expect that anthropometric characteristics affect external workload. However, we rather assume an inverted association, given that higher body mass leads to increased energy expenditure for given movements on sand, which might explain the differences between adolescent and adult male players. Lastly, the higher workload measured in adolescent players might also indicate an endeavor to compensate for techno-tactical deficits compared with more experienced adult players.

Zapardiel et al. were the first to perform conditional analyses in elite beach handball considering specific playing positions [19], and our data confirm the majority of their findings. For instance, specialists travel the greatest total distances per game and perform the most jumps per minute; goalkeepers perform the highest number of high-intensity accelerations; defenders complete high amounts of high-intensity decelerations and the most changes in direction; while wings reach the highest maximum velocities in the game, attain the highest maximum accelerations, and perform the highest amount of explosive efforts, defined as movements exceeding 3.5 m·s^−2^. These results indicate that physical demands in beach handball vary considerably according to the different playing positions. Hence, the availability of highly differentiated, valid, and reliable technical solutions to analyze external workload in sports using IMUs and LPS/ GPS should be complemented by differentiating more meaningfully within analyses according to playing positions. This will help in laying the basis for individualization when designing sound position-specific exercise programs and thus provides the foundation for optimal performance enhancement in already high-performing athletes.

### Limitations

The results of this study must be interpreted in light of some limitations. First, all measurements took place during tournaments in which we had to adhere to fixed schedules based on pre-determined match plans. Hence, no standardization was feasible regarding weather conditions such as temperature and/or humidity, which might have affected player performance and the condition of the sandy surface. We examined the effects of age and sex on external player load. To obtain a complete picture, the inclusion of a female adult group would have been desirable but, unfortunately, was not feasible to carry out within the funding period. Furthermore, the teams played up to four games on a given tournament day and up to seven games on a tournament weekend, which might have affected the outcomes as well. However, in sum, active playing time during seven games still falls below the mean active play time in male and female elite team handball in a given match [32]. The motivation of the players must be considered by acknowledging that their performance in third-place matches might deviate from the finals, which determines the tournament winner. Lastly, even though the validity of LPS was already demonstrated, limitations exist regarding data processing. As described in the Methods section, the Kalman filter was used to increase data accuracy. This algorithm is based on a linear dynamical system that suppresses rapid movement changes and may, in this context, lead to decreasing accuracy regarding fast CoD movements [7,21]. In addition, the algorithms used by manufacturers are frequently not explained in detail, so comparisons between systems should be made with caution [33].

## 5. Conclusions

Our results underline that beach handball can be characterized by numerous high-intensity actions such as sprints, jumps, changes in direction, as well as high-intensity accelerations and decelerations. In line with previous studies, we found moderate evidence of reduced external load during the second half of the matches, but contrary to previous reports, no interaction effects, which is indicative of overall fatigue instead of workload reductions affecting solely specific age/sex groups or playing positions. Our results further confirm studies demonstrating differences in external workload between female and male beach handball players but contradicted findings of higher workloads in adult compared to adolescent players. Particularly important information is derived from analyzing workload considering specific playing positions, which helps in designing individualized training programs for optimal performance enhancement in beach handball.

## Figures and Tables

**Figure 1 sensors-22-03011-f001:**
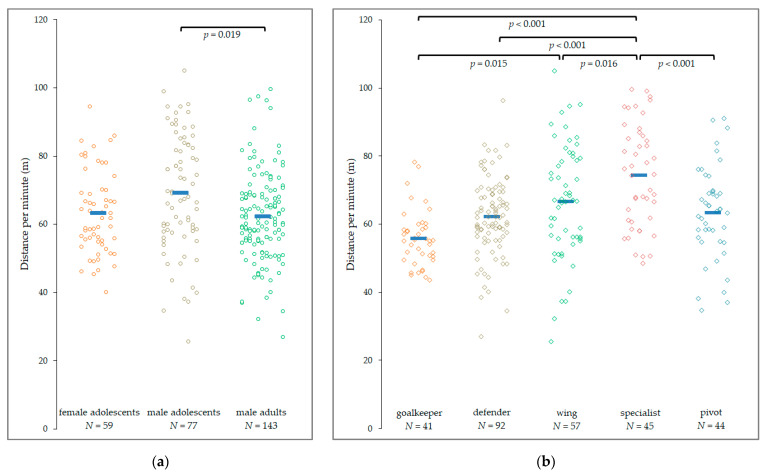
Distance per minute traveled across the subgroups: (**a**) subgroup 1 (age/sex); (**b**) subgroup 2 (playing position). Blue horizontal lines indicate the mean value.

**Figure 2 sensors-22-03011-f002:**
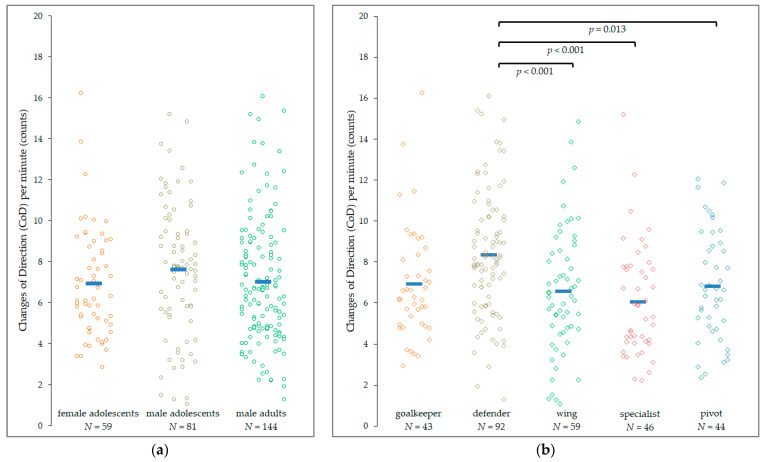
Number of changes in direction (counts/minute) across the subgroups: (**a**) subgroup 1 (age/sex); (**b**) subgroup 2 (playing position).

**Table 1 sensors-22-03011-t001:** Distribution and anthropometric characteristics of the total sample, and by group and playing position.

	*N*	Height (cm)	Body Mass (kg)	BMI (kg·m^−2^)
Total sample	69	187 ± 9	83.9 ± 11.5	23.9 ± 2.4
Subgroup 1: age/sex				
Female adolescents	16	176 ± 7	70.9 ± 7.9	22.9 ± 1.9
Male adolescents	19	188 ± 7	79.8 ± 6.6	22.5 ± 1.7
Male adults	34	190 ± 7	90.8 ± 9.7	25.1 ± 2.3
Subgroup 2: position				
Goalkeeper	12	186 ± 8	88.7 ± 12.9	25.7 ± 2.8
Defender	20	190 ± 9	87.7 ± 12.7	24.2 ± 1.9
Wing	11	192 ± 9	84.6 ± 8.6	22.9 ± 2.2
Specialist	10	184 ± 7	81.1 ± 12.2	23.9 ± 2.4
Pivot	16	184 ± 8	77.8 ± 8.0	23.1 ± 2.5

**Table 2 sensors-22-03011-t002:** Overview of statistical effects of age/sex and playing position on variables examined.

	Set		Subgroup 1		Subgroup 2	
	(1 vs. 2)		(Age/Sex)		(Playing Position)	
	*F*	*p*	*F*	*p* ^#^	*F*	*p* ^§^
D/min [m]	1.60	0.207	3.95	0.020	11.09	<0.001
V_MAX_ [km·h^−1^]	0.07	0.786	12.81	<0.001	3.91	0.004
PL/min [au]	4.86	0.028	20.05	<0.001	0.71	0.586
ACC_MAX_ [m·s^−2^]	0.02	0.879	15.60	<0.001	2.21	0.068
DEC_MAX_ [m·s^−2^]	0.02	0.883	6.11	0.003	4.85	<0.001
EE/min [counts]	0.86	0.355	1.92	0.149	4.67	0.001
ACC_IMA_/min [counts]	2.75	0.098	0.80	0.450	18.07	<0.001
DEC_IMA_/min [counts]	2.09	0.149	0.27	0.763	7.71	<0.001
CoD/min [counts]	14.36	<0.001	2.56	0.079	8.16	<0.001
Jumps/min [counts]	2.18	0.141	2.28	0.104	6.60	<0.001

^#^ Bonferroni-corrected for multiple comparisons and adjusted for playing position (subgroup 2 as covariate), ^§^ Bonferroni-corrected for multiple comparisons and adjusted for age/sex (subgroup 1 as covariate); D/min = distance traveled per minute (m), V_MAX_ = maximum velocity (km·h^−1^), PL/min = Player load per minute (au), EE/min = number of explosive efforts per minute, ACC_MAX_ = maximum acceleration (m·s^−2^), DEC_MAX_ = maximum deceleration (m·s^−2^), ACC_IMA_/min = number of IMA-detected accelerations per minute, DEC_IMA_/min = number of IMA-detected decelerations per minute, CoD/min = number of changes in direction per minute.

**Table 3 sensors-22-03011-t003:** Player load, maximum acceleration, and number of explosive efforts.

	PL/Min [au]	*N*	ACC_MAX_ [m·s^−2^]	*N*	EE/Min [Counts]	*N*
Sample	7.37 ± 1.96	283	2.97 ± 0.40	284	0.78 ± 0.48	283
Set 1	7.48 ± 2.15	282	2.76 ± 0.46	281	0.79 ± 0.61	283
Set 2	7.28 ± 2.10	280	2.76 ± 0.44	284	0.79 ± 0.57	281
Subgroup 1: age/sex						
Female adolescents	6.51 ± 1.24	59	2.75 ± 0.33	59	0.68 ± 0.44	59
Male adolescents	8.36 ± 2.24	79	3.08 ± 0.39	81	0.85 ± 0.57	80
Male adults	7.17 ± 1.82	145	3.01 ± 0.39	144	0.79 ± 0.44	144
Subgroup 2: position						
Goalkeeper	6.56 ± 1.67	44	2.89 ± 0.48	43	0.75 ± 0.38	42
Defender	7.45 ± 1.78	92	2.96 ± 0.36	92	0.65 ± 0.34	92
Wing	7.49 ± 2.20	58	3.07 ± 0.39	59	0.94 ± 0.60	59
Specialist	7.55 ± 2.21	45	2.90 ± 0.30	46	0.79 ± 0.54	46
Pivot	7.64 ± 1.88	44	3.03 ± 0.47	44	0.86 ± 0.53	44

PL/min = player load per minute (au), ACC_MAX_ = maximum acceleration (m/s^2^), EE/min = number of explosive efforts per minute (counts).

**Table 4 sensors-22-03011-t004:** IMA acceleration, IMA deceleration, and jumps.

	ACC_IMA_/Min [Counts]	*N*	DEC_IMA_/Min [Counts]	*N*	Jumps/Min [Counts]	*N*
Sample	1.54 ± 0.69	284	1.70 ± 0.80	284	0.94 ± 0.68	278
Set 1	1.60 ± 0.86	282	1.75 ± 0.96	281	1.00 ± 0.81	277
Set 2	1.51 ± 0.76	281	1.67 ± 0.89	278	0.92 ± 0.74	275
Subgroup 1: age/sex						
Female adolescents	1.47 ± 0.74	59	1.63 ± 0.53	59	0.79 ± 0.47	56
Male adolescents	1.64 ± 0.74	81	1.68 ± 0.82	81	1.06 ± 0.62	81
Male adults	1.52 ± 0.64	144	1.74 ± 0.87	144	0.93 ± 0.76	141
Subgroup 2: position						
Goalkeeper	2.11 ± 0.62	44	1.44 ± 0.55	43	0.48 ± 0.36	39
Defender	1.36 ± 0.52	91	2.03 ± 0.83	92	0.88 ± 0.66	91
Wing	1.56 ± 0.77	59	1.56 ± 0.85	59	1.02 ± 0.75	59
Specialist	1.12 ± 0.59	46	1.36 ± 0.68	46	1.20 ± 0.71	45
Pivot	1.78 ± 0.59	44	1.82 ± 0.70	44	1.11 ± 0.57	44

ACC_IMA_/min = number of IMA-detected accelerations per minute, DEC_IMA_/min = number of IMA-detected decelerations per minute.

## Data Availability

The data represented in the study are openly available in OSF Framework (https://doi.org/10.17605/OSF.IO/ZC9R2).

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
