# Peer review of "External Load Analysis in Beach Handball Using a Local Positioning System and Inertial Measurement Units"

_sensors, 2022, doi:10.3390/s22083011_

Round 1

Reviewer 1 Report

as above for authors;

sensors-1679162

The paper covers interesting information on beach handball, which is seldom characterized as a sports discipline. The paper is clear and well written. Some comments are attached below:

L 24 Keywords: please exclude the word “catapult” from the keyword, it is not a clear keyword, but a producer name;

L 110 perspective team – it is not clear what do you mean

L 114 does it mean that they agreed? Do you have written permission?

L 115 please give the number of decision

L 147 please add the legend – what is "a" should be given

L 196 keep the same names of groups throughout the paper 

L 281 and 287 – table 2 and 3?

L 321- 328 – it seems repetition of the text written in the introduction. Not necessary.

L 381, 383 – U16, 18 should be given at least the first time in the full name. For the wider audience, it can be not clear enough.

L 417 – 5-m acceleration – please do not use such shortcuts- 5m distance for acceleration/ acceleration distance?

L 446-449 – do you mind people's preferences/compatibilities here?

L 464 – 467 – that looks like the introduction, not a conclusion from your research. Conclusions should come exactly from your research without common sentences. 

It seems that you have omitted following paper, that can be of the great interest for you:

Iannaccone, Alice, et al. "Relationship Between External and Internal Load Measures in Youth Beach Handball." International journal of sports physiology and performance 17.2 (2021): 256-262.

Reviewer 2 Report

Thanks for the opportunity to review the manuscript titled, " External Load Analysis in German Beach-Handball Using a  Local Positioning System and Inertial Measurement Units". The study aims to evaluate the external load between the first set and second of a game and explore the influence of sex/age and positions on external loads in elite Beach-Handball players during competitions. The study makes use of the local positioning system and a single IMU as a measurement tool. Which I think fits the scope of the special issue in Sensors. In general, the manuscript is well written, and the study seems well planned. The study demonstrates a use case of using portable sensors to study the players' performance in a real-world situation. The study could facilitate the development of individualised training programs for elite athletes. I have no major concern regarding the quality of the study.

Minor comments / Suggestion

Title:

  1. I think not necessary to mention the country where the study was conducted in the title.

Introduction:

  1. Is there any break in between the two half of a game?

Methods

  1. Please provide more technical details of the LPS system. For example the, precision and reliability.
  2. How does a 'Jump' is identified from the IMU data?
  3. For the definition of CoD, do you mean that the player moved from '45° to 135° to the right' to '-45° to -135° to the left'? (Line 153 – 154)

Discussion:

  1. There is no adult female enrolled in this study, and this should be one of the limitations of the study.
  2. Given that the player positions are the most important factor for estimating external loading. I would suggest the authors include scatter plots similar to the Figures 1&2b for each measure in the supplementary materials. Thus the reader can have a better understanding of the differences between positions.

Other:

The data DOI: 10.17605/OSF.IO/ZC9R2 is not available at the time of review.

Reviewer 3 Report

Thank you for the opportunity to review this manuscript.

Congratulations for your study and effort. However, I would like you to pay attention to the indications atached for improvement its importance.   
